# Temporal Positive-unlabeled Learning for Biomedical Hypothesis Generation via Risk Estimation

Uchenna Akujuobi[1,2]     Jun Chen[1]     Mohamed Elhoseiny[1]
Michael Spranger[2]     Xiangliang Zhang[1 ✉]
[1]King Abdullah University of Science and Technology     [2] Sony AI, Tokyo
{uchenna.akujuobi,jun.chen,mohamed.elhoseiny,xiangliang.zhang}@kaust.edu.sa
michael.spranger@gmail.com

## Abstract

Understanding the relationships between biomedical terms like viruses, drugs, and symptoms is essential in the fight against diseases. Many attempts have been made to introduce the use of machine learning to the scientific process of *hypothesis generation* (HG), which refers to the discovery of meaningful implicit *connections* between biomedical terms. However, most existing methods fail to truly capture the *temporal dynamics of scientific term relations* and also assume unobserved connections to be irrelevant (i.e., in a positive-negative (PN) learning setting). To break these limits, we formulate this HG problem as future connectivity prediction task on a dynamic attributed graph via positive-unlabeled (PU) learning. Then, the key is to capture the temporal evolution of node pair (term pair) relations from just the positive and unlabeled data. We propose a variational inference model to estimate the positive prior, and incorporate it in the learning of node pair embeddings, which are then used for link prediction. Experiment results on real-world biomedical term relationship datasets and case study analyses on a COVID-19 dataset validate the effectiveness of the proposed model.

## 1   Introduction

Recently, the study of co-relationships between biomedical entities is increasingly gaining attention. The ability to predict future relationships between biomedical entities like diseases, drugs, and genes enhances the chances of early detection of disease outbreaks and reduces the time required to detect probable disease characteristics. For instance, in 2020, the COVID-19 outbreak pushed the world to a halt with scientists working tediously to study the disease characteristics for containment, cure, and vaccine. An increasing number of articles encompassing new knowledge and discoveries from these studies were being published daily [1]. However, with the accelerated growth rate of publications, the manual process of reading to extract undiscovered knowledge increasingly becomes a tedious and time-consuming task beyond the capability of individual researchers.

In an effort towards an advanced knowledge discovery process, computers have been introduced to play an ever-greater role in the scientific process with automatic hypothesis generation (HG). The study of automated HG has attracted considerable attention in recent years [41, 25, 45, 47]. Several previous works proposed techniques based on association rules [25, 18, 47], clustering and topic modeling [45, 44, 5], text mining [43, 42], and others [28, 49, 39]. However, these previous works fail to truly utilize the crucial information encapsulated in the dynamic nature of scientific discoveries and assume that the unobserved relationships denote a non-relevant relationship (*negative*).

To model the historical evolution of term pair relations, we formulate HG on a term relationship graph $G = \{V, E\}$, which is decomposed into a sequence of attributed graphlets $G = \{G^1, G^2, ..., G^T\}$, where the graphlet at time $t$ is defined as,

**Definition 1. Temporal graphlet**: *A temporal graphlet $G^t = \{V^t, E^t, x_v^t\}$ is a temporal subgraph at time step t, which consists of nodes (terms) $V^t$ satisfying $V^1 \subseteq V^2, ..., \subseteq V^T$ and the observed co-occurrence between these terms $E^t$ satisfying $E^1 \subseteq E^2, ..., \subseteq E^T$. And $x_v^t$ is the node attribute.*

Example of the node terms can be *covid-19, fever, cough, Zinc, hepatitis B virus* etc. When two terms co-occurred at time $t$ in scientific discovery, a link between them is added to $E^t$, and the nodes are added to $V^t$ if they haven't been added.

**Definition 2. Hypothesis Generation** *(HG): Given $G = \{G^1, G^2, ..., G^T\}$, the target is to predict which nodes unlinked in $V^T$ should be linked (a hypothesis is generated between these nodes).*

We address the HG problem by modeling how $E^t$ was formed from $t = 1$ to $T$ (on a dynamic graph), rather than using only $E^T$ (on a static graph). In the design of learning model, it is clear to us the observed edges are *positive*. However, we are in a dilemma whether the unobserved edges are *positive* or *negative*. The prior work simply set them to be *negative*, learning in a positive-negative (PN) setting) based on a closed world assumption that unobserved connections are irrelevant (*negative*) [39, 28, 4]. We set the learning with a more realistic assumption that the unobserved connections are a mixture of positive and negative term relations (*unlabeled*), a.k.a. Positive-unlabeled (PU) learning, which is different from semi-supervised PN learning that assumes a known set of labeled negative samples. For the observed positive samples in PU learning, they are assumed to be selected entirely at random from the set of all positive examples [16]. This assumption facilitates and simplifies both theoretical analysis and algorithmic design since the probability of observing the label of a positive example is constant. However, estimating this probability value from the positive-unlabeled data is nontrivial. We propose a variational inference model to estimate the positive prior and incorporate it in the learning of node pair embeddings, which are then used for link prediction (hypothesis generation).

We highlight the contributions of this work as follows.

1) Methodology: we propose *a PU learning approach on temporal graphs*. It differs from other existing approaches that learn in a conventional PN setting on static graphs. In addition, we estimate the positive prior via a variational inference model, rather than setting by prior knowledge.

2) Application: to the best of our knowledge, this is the first the application of PU learning on the HG problem, and on dynamic graphs. We applied the proposed model on real-world graphs of terms in scholarly publications published from 1945 to 2020. Each of the three graphs has around 30K nodes and 1-2 million edges. The model is trained end-to-end and shows superior performance on HG. Case studies demonstrate our new and valid findings of the positive relationship between medical terms, including newly observed terms that were not observed in training.

## 2  Related Work of PU Learning

In PU learning, since the negative samples are not available, a classifier is trained to minimize the expected misclassification rate for both the positive and unlabeled samples. One group of study [32, 31, 33, 22] proposed a two-step solution: 1) identifying reliable negative samples, and 2) learning a classifier based on the labeled positives and reliable negatives using a (semi)-supervised technique. Another group of studies [36, 30, 26, 17, 40] considered the unlabeled samples as negatives with label noise. Hence, they place higher penalties on misclassified positive examples or tune a hyperparameter based on suitable PU evaluation metrics. Such a proposed framework follows the SCAR (Selected Completely at Random) assumption since the noise for negative samples is constant.

**PU Learning via Risk Estimation**  Recently, the use of unbiased risk estimator has gained attention [12, 14, 15, 48]. The goal is to minimize the expected classification risk to obtain an empirical risk minimizer. Given an input representation $h$ (in our case the node pair representation to be learned), let $f : \mathbb{R}^d \to \mathbb{R}$ be an arbitrary decision function and $l : \mathbb{R} \times \{\pm 1\} \to \mathbb{R}$ be the loss function calculating the incurred *loss* $l(f(h), y)$ of predicting an output $f(h)$ when the true value is $y$. Function $l$ has a variety of forms, and is determined by application needs [29, 13]. In PN learning, the empirical risk minimizer $\hat{f}_{PN}$ is obtained by minimizing the PN risk $\hat{\mathcal{R}}(f)$ w.r.t. a class prior of $\pi_p$:

$$\hat{\mathcal{R}}(f) = \pi_P \hat{\mathcal{R}}_P^+(f) + \pi_N \hat{\mathcal{R}}_N^-(f), \tag{1}$$

where $\pi_N = 1 - \pi_P$, $\hat{\mathcal{R}}_P^+(f) = \frac{1}{n_P} \sum_{i=1}^{n_P} l(f(h_i^P), +1)$ and $\hat{\mathcal{R}}_N^-(f) = \frac{1}{n_N} \sum_{i=1}^{n_N} l(f(h_i^N), -1)$. The variables $n_P$ and $n_N$ are the numbers of positive and negative samples, respectively.

PU learning has to exploit the fact that $\pi_N p_N(h) = p(h) - \pi_P p_P(h)$, due to the absence of negative samples. The second part of Eq. (1) can be reformulated as:

$$\pi_N \hat{\mathcal{R}}_N^-(f) = \hat{\mathcal{R}}_U^- - \pi_P \hat{\mathcal{R}}_P^-(f), \tag{2}$$

where $\mathcal{R}_U^- = \mathbb{E}_{h \sim p(h)}[l(f(h), -1)]$ and $\mathcal{R}_P^- = \mathbb{E}_{h \sim p(h|y=+1)}[l(f(h), -1)]$. Furthermore, the classification risk can then be approximated by:

$$\hat{\mathcal{R}}_{PU}(f) = \pi_P \hat{\mathcal{R}}_P^+(f) + \hat{\mathcal{R}}_U^-(f) - \pi_P \hat{\mathcal{R}}_P^-(f), \tag{3}$$

where $\hat{\mathcal{R}}_P^-(f) = \frac{1}{n_P} \sum_{i=1}^{n_P} l(f(h_i^P), -1)$, $\hat{\mathcal{R}}_U^-(f) = \frac{1}{n_U} \sum_{i=1}^{n_U} l(f(h_i^U), -1)$, and $n_U$ is the number of unlabeled data sample. To obtain an empirical risk minimizer $\hat{f}_{PU}$ for the PU learning framework, $\hat{\mathcal{R}}_{PU}(f)$ needs to be minimized. Kiryo et al. noted that the model tends to suffer from overfitting on the training data when the model $f$ is made too flexible [29]. To alleviate this problem, the authors proposed the use of non-negative risk estimator for PU learning:

$$\tilde{\mathcal{R}}_{PU}(f) = \pi_P \hat{\mathcal{R}}_P^+(f) + \max\{0, \hat{\mathcal{R}}_U^- - \pi_P \hat{\mathcal{R}}_P^-(f)\}. \tag{4}$$

It works in fact by explicitly constraining the training risk of PU to be non-negative. The key challenge in practical PU learning is the unknown of prior $\pi_P$.

**Prior Estimation** The knowledge of the class prior $\pi_P$ is quintessential to estimating the classification risk. In PU learning for our node pairs, we represent a sample as $\{h, s, y\}$, where $h$ is the node pair representation (to be learned), $s$ indicates if the pair relationship is observed (labeled, $s = 1$) or unobserved (unlabeled, $s = 0$), and $y$ denotes the true class (positive or negative). We have only the positive samples labeled: $p(y = 1|s = 1) = 1$. If $s = 0$, the sample can belong to either the positive or negative class. PU learning runs commonly with the Selected Completely at Random (SCAR) assumption, which postulates that the labeled sample set is a random subset of the positive sample set [16, 6, 8]. The probability of selecting a positive sample to observe can be denoted as: $p(s = 1|y = 1, h)$. The SCAR assumption means: $p(s = 1|y = 1, h) = p(s = 1|y = 1)$. However, it is hard to estimate $\pi_P = p(y = 1)$ with only a small set of observed samples ($s = 1$) and a large set of unobserved samples ($s = 0$) [7]. Solutions have been tried by i) estimating from a validation set of a fully labeled data set (all with $s = 1$ and knowing $y = 1$ or $-1$) [29, 10]; ii) estimating from the background knowledge; and iii) estimating directly from the PU data [16, 6, 8, 27, 14]. In this paper, we focus on estimating the prior directly from the PU data. Specifically, unlike the other methods, we propose a scalable method based on deep variational inference to jointly estimate the prior and train the classification model end-to-end. The proposed deep variational inference uses KL-divergence to estimate the parameters of class mixture model distributions of the positive and negative class in contrast to the method proposed in [14] which uses penalized L1 divergences to assign higher penalties to class priors that scale the positive distribution as more than the total distribution.

# 3 PU learning on Temporal Attributed Networks

## 3.1 Model Design

The architecture of our Temporal Relationship Predictor (TRP) model is shown in Fig. 1. For a given pair of nodes $a^{ij} = <v_i, v_j>$ in any temporal graphlet $G^t$, the main steps used in the training process of TRP for calculating the connectivity prediction score $p^t(a^{ij})$ are given in Algorithm 1. The testing process also uses the same Algorithm 1 (with t=T), calculating $p^T(a^{ij})$ for *node pairs that have not been connected in* $G^{T-1}$. The connectivity prediction score is calculated in line 6 of Algorithm 1 by $p^t(a^{ij}) = f_C(h_{a^{ij}}^t; \theta_C)$, where $\theta_C$ is the classification network parameter, and the embedding vector $h_{a^{ij}}^t$ for the pair $a^{ij}$ is iteratively updated in lines 1-5. These iterations of updating $h_{a^{ij}}^t$ are shown as the recurrent structure in Fig. 1 (a), followed by the classifier $f_C(.; \theta_C)$.

The recurrent update function $h_{a^{ij}}^\tau = f_A(h_{a^{ij}}^{\tau-1}, z_{v_i}^\tau, z_{v_j}^\tau; \theta_A)$, $\tau = 1...t$, in line 4 is shown in Fig. 1 (b), and has a Gated recurrent unit (GRU) network at its core,

$$\mathcal{P} = \sigma_g(W^z f_m(z_{v_i}^\tau, z_{v_j}^\tau) + U^{\mathcal{P}} h_{a^{ij}}^{\tau-1} + b^{\mathcal{P}}),$$
$$r = \sigma_g(W^r f_m(z_{v_i}^\tau, z_{v_j}^\tau) + U^r h_{a^{ij}}^{\tau-1} + b^r),$$
$$\tilde{h}_{a^{ij}}^\tau = \sigma_{h'}(W f_m(z_{v_i}^\tau, z_{v_j}^\tau) + r \circ U h_{a^{ij}}^{\tau-1} + b),$$
$$h_{a^{ij}}^\tau = \mathcal{P} \circ \tilde{h}_{a^{ij}}^\tau + (1 - \mathcal{P}) \circ h_{a^{ij}}^{\tau-1}. \tag{5}$$

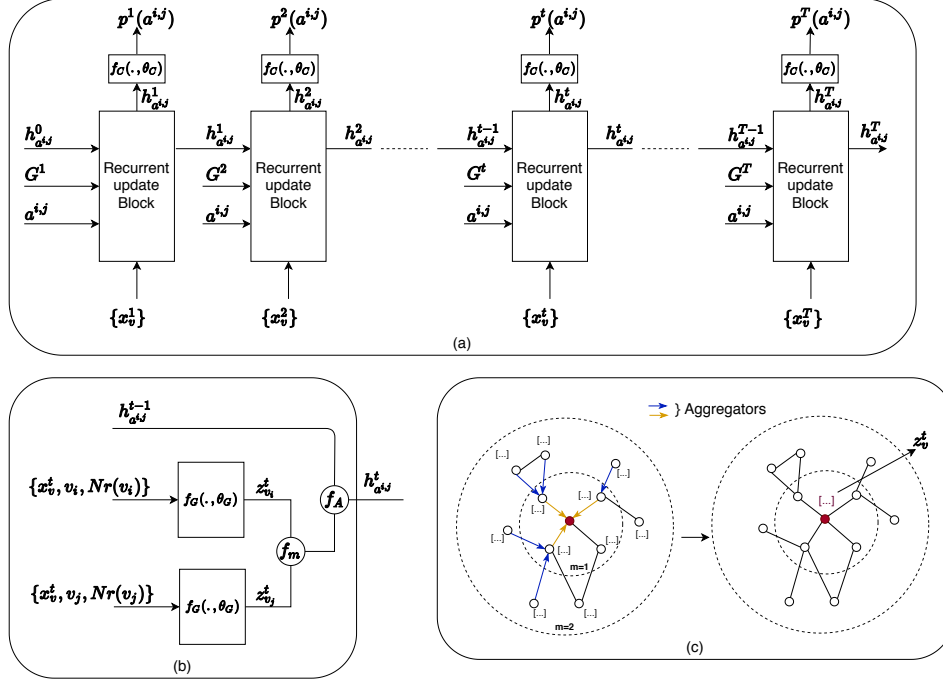

Figure 1: The proposed TRP model. Block (a) shows the outer view of the model framework. The inner structure of the recurrent update block and neighborhood aggregation method are shown in block (b) and (c), respectively.

---

**Algorithm 1:** Calculate the future connection score for term pairs $a^{i,j} = < v_i, v_j >$

---

**Input:** $G = \{G^1, G^2, \ldots, G^T\}$ with node feature $x_v^t$, a node pair $a^{i,j} = < v_i, v_j >$ in $G^t$, and an initialized pair embedding vector $h_{a^{i,j}}^0$ (e.g., by zeros)

**Result:** $p_{a^{i,j}}^t$, the connectivity prediction score for the node pair $a^{i,j}$

1   **for** $\tau \leftarrow 1 \cdots t$ **do**
2      Obtain the current node feature $x_v^\tau$ ($v = v_i, v_j$) of both nodes (terms) $v_i, v_j$; as well as $x_{Nr(v)}^\tau$
      ($v = v_i, v_j$) for the node feature of sampled neighboring nodes for $v_i, v_j$;
3      Aggregate the neighborhood information of node $v = v_i, v_j$, $z_v^\tau = f_G(x_v^\tau, x_{Nr(v)}^\tau; \theta_G)$;
4      Update the embedding vector for the node pair $h_{a^{i,j}}^\tau = f_A(h_{a^{i,j}}^{\tau-1}, z_{v_i}^\tau, z_{v_j}^\tau; \theta_A)$ ;
5   **end**
6   Return $p_{a^{i,j}}^t = f_C(h_{a^{i,j}}^t; \theta_C)$

---

where $\circ$ denotes element-wise multiplication, $\sigma$ is a nonlinear activation function, and $f_m(.)$ is an aggregation function. In this study, we use a max pool aggregation. The variables $\{W, U\}$ are the weights. The inputs to function $f_A$ include: $h_{a^{ij}}^{\tau-1}$, the embedding vector in previous step; $\{z_{v_i}^t, z_{v_j}^t\}$, the representation of node $v_i$ and $v_j$ after aggregation their neighborhood, $z_v^\tau = f_G(x_v^\tau, x_{Nr(v)}^\tau; \theta_G)$, given in line 3. The aggregation function $f_G$ takes input the node feature $x_v^\tau$, and the neighboring node feature $x_{Nr(v)}^\tau$ and goes through the aggregation block shown in Fig. 1 (c). The aggregation network $f_G(; \theta_G)$ is implemented following GraphSAGE [21], which is one of the most popular graph neural networks for aggregating node and its neighbors.

The loss function in our problem $l(p^t(a^{ij}), y)$ evaluates the loss incurred by predicting a connectivity $p^t(a^{ij}) = f_C(h_{a^{ij}}^t; \theta_C)$ when the ground truth is $y$. For constructing the training set for our PU learning in the dynamic graph, we first clarify the label notations. For one pair $a^{ij}$ from a graph $G^t$, its label $y_t^{ij} = 1$ (*positive*) if the two nodes have a link observed in $G^{t+1}$ (they have an edge $\in E^{t+1}$, observed in next time step), i.e., $s_t^{ij} = 1$. Otherwise when no link is observed between them in $G^{t+1}$, $a^{ij}$ is *unlabeled*, i.e., $s_t^{ij} = 0$, since $y_t^{ij}$ can be either 1 or -1. Since we consider insertion

only graphlets sequence, $V^1 \subseteq V^2, ..., \subseteq V^T$ and $E^1 \subseteq E^2, ..., \subseteq E^T$, $y_t^{ij} = 1$ maintains for all future steps after $t$ (once positive, always positive). At the final step $t = T$, all pairs with observed connections already have $y_T^{ij} = 1$, our objective is to predict the connectivity score for those pairs with $s_T^{ij} = 0$. Our loss function is defined following the unbiased risk estimator in Eq. (3),

$$\mathcal{L}^R = \pi_P \hat{\mathcal{R}}_P^+(f_C) + \hat{\mathcal{R}}_U^-(f_C) - \pi_P \hat{\mathcal{R}}_P^-(f_C), \tag{6}$$

where $\hat{\mathcal{R}}_P^+(f_C) = \frac{1}{|\mathfrak{H}_P|} \sum_{a^{ij} \in \mathfrak{H}_P} 1/(1 + exp(p^t(a^{ij})))$, $\hat{\mathcal{R}}_U^-(f_C) = \frac{1}{|\mathfrak{H}_U|} \sum_{a^{ij} \in \mathfrak{H}_U} 1/(1 + exp(-p^t(a^{ij})))$, and $\hat{\mathcal{R}}_P^-(f_C) = \frac{1}{|\mathfrak{H}_P|} \sum_{a^{ij} \in \mathfrak{H}_P} 1/(1 + exp(-p^t(a^{ij})))$ with the positive samples $\mathfrak{H}_P$ and unlabeled samples $\mathfrak{H}_U$, when taking $l$ as sigmoid loss function. $\mathcal{L}^R$ can be adjusted with the non-negative constraint in Eq. (4), with the same definition of $\hat{\mathcal{R}}_P^+(f)$, $\hat{\mathcal{R}}_U^-$, and $\hat{\mathcal{R}}_P^-(f)$.

## 3.2 Prior Estimation

The positive prior $\pi_P$ is a key factor in $\mathcal{L}^R$ to be addressed. The samples we have from $G$ are only positive $\mathfrak{H}_P$ and unlabeled $\mathfrak{H}_U$. Due to the absence of negative samples and of prior knowledge, we present an estimate of the class prior from the distribution of $h$, which is the pair embedding from $f_A$. Without loss of generality, we assume that the learned $h$ of all samples has a Gaussian mixture distribution of two components, one is for the positive samples, while the other is for the negative samples although they are unlabeled. The mixture distribution is parameterized by $\beta$, including the mean, co-variance matrix and mixing coefficient of each component. We learn the mixture distribution using stochastic variational inference [24] via the "Bayes by Backprop" technique [9]. The use of variational inference has been shown to have the ability to model salient properties of the data generation mechanism and avoid singularities. The idea is to find variational distribution variables $\theta^*$ that minimizes the Kullback-Leibler (KL) divergence between the variational distribution $q(\beta|\theta)$ and the true posterior distribution $p(\beta|h)$:

$$\theta^* = \arg \min_\theta \mathcal{L}^E, \tag{7}$$
$$\text{where,} \quad \mathcal{L}^E = KL(q(\beta|\theta)||p(\beta|h)) = KL(q(\beta|\theta)||p(\beta)) - \mathbb{E}_{q(\beta|\theta)}[\log p(h|\beta)].$$

The resulting cost function $\mathcal{L}^E$ on the right of Eq. (7) is known as the (negative) "evidence lower bound" (ELBO). The second term in $\mathcal{L}^E$ is the likelihood of $h$ fitting to the mixture Gaussian with parameter $\beta$: $\mathbb{E}_{q(\beta|\theta)}[\log p(h|\beta)]$, while the first term is referred to as the complexity cost [9]. We optimize the ELBO using stochastic gradient descent. With $\theta^*$, the positive prior is then estimated as

$$\pi_P = q(\beta_i^\pi | \theta^*), i = \arg \max_{k=1,2} |C_k| \tag{8}$$

where $C_1 = \{h \in \mathfrak{H}_P, p(h|\beta^1) > p(h|\beta^2)\}$ and $C_2 = \{h \in \mathfrak{H}_P, p(h|\beta^2) > p(h|\beta^1)\}$.

## 3.3 Parameter Learning

To train the three networks $f_A(.;\theta_A)$, $f_G(.;\theta_G)$, $f_C(.;\theta_C)$ for connectivity score prediction, we jointly optimize $\mathcal{L} = \sum_{t=1}^T \mathcal{L}_t^R + \mathcal{L}_t^E$, using Adam over the model parameters. Loss $L^R$ is the PU classification risk as described in section 3.1, and $L^E$ is the loss of prior estimation as described in section 3.2. Note that during training, $y_{a^{ij}}^T = y_{a^{ij}}^{T-1}$ since we do not observe $G^T$ in training. This is to enforce prediction consistency.

# 4 Experimental Evaluation

## 4.1 Dataset and Experimental Setup

The graphs on which we apply our model are constructed from the title and abstract of papers published in the biomedical fields from 1949 to 2020. The nodes are the biomedical terms, while the edges linking two nodes indicate the co-occurrence of the two terms. Note that we focus only on the co-occurrence relation and leave the polarity of the relationships for future study. To evaluate the model's adaptivity in different scientific domains, we construct three graphs from papers relevant to *COVID-19*, *Immunotherapy*, and *Virology*. The graph statistics are shown in Table 1. To set up the

training and testing data for TRP model, we split the graph by year intervals (5 years for *COVID-19* or 10 years for *Virology* and *Immunotherapy*). We use splits of $\{G^1, G^2, ..., G^{T-1}\}$ for training, and use connections newly added in the final split $G^T$ for testing. Since baseline models do not work on dynamic data, hence we train on $G^{T-1}$ and test on new observations made in $G^T$. Therefore in testing, the *positive* pairs are those linked in $G^T$ but not in $G^{T-1}$, i.e., $E^T \setminus E^{T-1}$, which can be new connections between nodes already existing in $G^{T-1}$, or between a new node in $G^T$ and another node in $G^{T-1}$, or between two new nodes in $G^T$. All other unlinked node pairs in $G^T$ are *unlabeled*.

At each $t = 1, ..., T-1$, graph $G^t$ is incrementally updated from $G^{t-1}$ by adding new nodes (biomedical terms) and their links. For the node feature vector $x_v^t$, we extract its term description and convert to a 300-dimensional feature vector by applying the latent semantic analysis (LSI). The missing term and context attributes are filled with zero vectors. If this node already exists before time $t$, the context features are updated with the new information about them in discoveries, and publications. In the inference (testing) stage, the new nodes in $G^T$ are only presented with their feature vectors $x_v^T$. The connections to these isolated nodes are predicted by our TRP model.

We implement TRP using the Tensorflow library. Each GPU based experiment was conducted on an Nvidia 1080TI GPU. In all our experiments, we set the hidden dimensions to $d = 128$. For each neural network based model, we performed a grid search over the learning rate $lr = \{1e-2, 5e-3, 1e-3, 5e-2\}$, For the prior estimation, we adopted Gaussian, square-root inverse Gamma, and Dirichlet distributions to model the mean, co-variance matrix and mixing coefficient variational posteriors respectively.

## 4.2 Comparison Methods and Performance Matrices

We evaluate our proposed TRP model in several variants and by comparing with several competitors:

1) **TRP variants**: a) *TRP-PN* - the same framework but in PN setting (i.e., treating all unobserved samples as *negative*, rather than *unlabeled*); b) *TRP-nnPU* - trained using the non-negative risk estimator Eq. (4) or the equation defined in section 3.1 for our problem; and c) *TRP-uPU* - trained using the unbiased PU risk estimator Eq. (3), the equation defined in section 3.1 for our problem. The comparison of these variants will show the impact of different risk estimators.

2) **SOTA PU learning**: the state-of-the-art (SOTA) PU learning methods taking input $h$ from the SOTA node embedding models, which can be based on LSI [11], node2vec [20], DynAE [19] and GraphSAGE [21]. Since node2vec learns only from the graph structure, we concatenate the node2vec embeddings with the text (term and context) attributes to obtain an enriched node representation. Unlike our TRP that learns $h$ for one pair of nodes, these models learn embedding vectors for individual nodes. Then, $h$ of one pair from baselines is defined as the concatenation of the embedding vector of two nodes. We observe from the results that a concatenation of node2vec and LSI embeddings had the most competitive performance compared to others. Hence we only report the results based on concatenated embeddings for all the baselines methods. The used SOTA PU learning methods include [16] by reweighting all examples, and models with different estimation of the class prior such as SAR-EM [8] (an EM-based SAR-PU method), SCAR-KM2 [38], SCAR-C [8], SCAR-TIcE [6], and pen-L1 [14].

3) **Supervised:** weaker but simpler logistic regression applied also $h$.

We measure the performance using four different metrics. These metrics are the Macro-F1 score (F1-M), F1 score of observed connections (F1-S), F1 score adapted to PU learning (F1-P) [7, 30], and the label ranking average precision score (LRAP), where the goal is to give better rank to the positive node pairs. In all metrics used, higher values are preferred.

## 4.3 Evaluation Results

Table 2 shows that TRP-uPU always has the superior performance over all other baselines across the datasets due to its ability to capture and utilize temporal, structural, and textual information (learning better $h$) and also the better class prior estimator. Among TRP variants, TRP-uPU has higher or equal F1 values comparing to the other two, indicating the benefit of using the unbiased risk estimator. On the LRAP scores, TRP-uPU and TRP-PN have the same performance on promoting the rank of the positive samples. Note that the results in Table 2 are from the models trained with their best learning rate, which is an important parameter that should be tuned in gradient-based optimizer, by either exhaustive search or advanced auto-machine learning [35]. To further investigate the performance of

Table 1: Three graph dataset statistics, with their number of nodes and edges

| | Graphs until $T$ | | Node pairs in evaluation at $T$ | |
|---|---|---|---|---|
| | #nodes | #edges | #Positive $^{E^T \setminus E^{T-1}}$ | #Unlabeled $^{\text{sampled from } \{V^T \times V^T\} \setminus E^T}$ |
| COVID-19 | 27,325 | 2,474,624 | 655,649 | 1,019,458 |
| Immunotherapy | 28,823 | 919,004 | 303,516 | 1,075,659 |
| Virology | 38,956 | 1,117,118 | 446,574 | 1,382,856 |

Table 2: Evaluation results on the COVID-19, Immuniotherapy and Neurology datasets, respectively

| | COVID-19 | | | | Virology | | | | Immunotherapy | | | |
|---|---|---|---|---|---|---|---|---|---|---|---|---|
| | F1-S | F1-M | F1-P | LRAP | F1-S | F1-M | F1-P | LRAP | F1-S | F1-M | F1-P | LRAP |
| Supervised | 0.82 | 0.86 | 1.73 | 0.77 | 0.57 | 0.73 | 1.42 | 0.43 | 0.67 | 0.80 | 2.18 | 0.56 |
| SCAR-C [8] | 0.82 | 0.86 | 1.73 | 0.77 | 0.56 | 0.72 | 1.40 | 0.43 | 0.66 | 0.79 | 2.14 | 0.56 |
| SCAR-KM2 [38] | 0.76 | 0.82 | 1.52 | 0.73 | 0.49 | 0.61 | 1.21 | 0.33 | 0.53 | 0.66 | 1.50 | 0.36 |
| SCAR-TIcE [6] | 0.57 | 0.30 | 1.01 | 0.39 | 0.38 | 0.22 | 1.02 | 0.23 | 0.35 | 0.19 | 1.01 | 0.22 |
| SAR-EM [6] | 0.78 | 0.83 | 1.68 | 0.76 | 0.60 | 0.75 | 1.52 | 0.60 | 0.67 | 0.80 | 2.18 | 0.62 |
| Elkan [16] | 0.82 | 0.86 | 1.74 | 0.81 | 0.58 | 0.73 | 1.47 | 0.58 | 0.69 | 0.81 | 2.26 | 0.65 |
| TRP-PN | 0.84 | 0.87 | 1.80 | **0.91** | 0.73 | **0.83** | 2.31 | 0.81 | **0.71** | 0.81 | 2.33 | **0.77** |
| penL1-nnPU | 0.71 | 0.70 | 1.35 | 0.45 | 0.61 | 0.71 | 1.83 | 0.63 | 0.53 | 0.63 | 1.62 | 0.73 |
| TRP-nnPU | 0.80 | 0.82 | 1.68 | 0.89 | 0.73 | 0.82 | 2.38 | 0.83 | 0.67 | 0.78 | 2.18 | 0.76 |
| penL1-uPU | 0.85 | **0.88** | 1.86 | 0.89 | **0.74** | **0.83** | **2.45** | **0.81** | 0.70 | 0.81 | 2.33 | 0.72 |
| TRP-uPU | **0.86** | **0.88** | **1.88** | **0.91** | **0.74** | **0.83** | 2.38 | **0.81** | **0.71** | **0.82** | **2.35** | **0.77** |

TRP variants, we show in Figure 2 their F1-S at different learning rate in trained from 1 to 10 epochs. We notice that TRP-uPU has a stable performance across different epochs and learning rates. This advantage is attributed to the unbiased PU risk estimation, which learns from only positive samples with no assumptions on the negative samples. We also found interesting that nnPU was worse than uPU in our experimental results. However, it is not uncommon for uPU to outperform nnPU in evaluation with real-world datasets. Similar observations were found in the results in [14, 17]. In our case, we attribute this observation to the joint optimization of the loss from the classifier and the prior estimation. Specifically, in the loss of uPU (Eq. (3)), $\pi_P$ affects both $\hat{\mathcal{R}}_P^+(f)$ and $\hat{\mathcal{R}}_P^-(f)$. However, in the loss of nnPU (Eq. (4)), $\pi_P$ only weighted $\hat{\mathcal{R}}_P^+(f)$ when $\hat{\mathcal{R}}_U^- - \pi_P\hat{\mathcal{R}}_P^-(f)$ is negative. In real-world applications, especially when the true prior is unknown, the loss selection affects the estimation of $\pi_P$, and thus the final classification results. TRP-PN is not as stable as TRP-uPU due to the strict assumption of unobserved samples as negative.

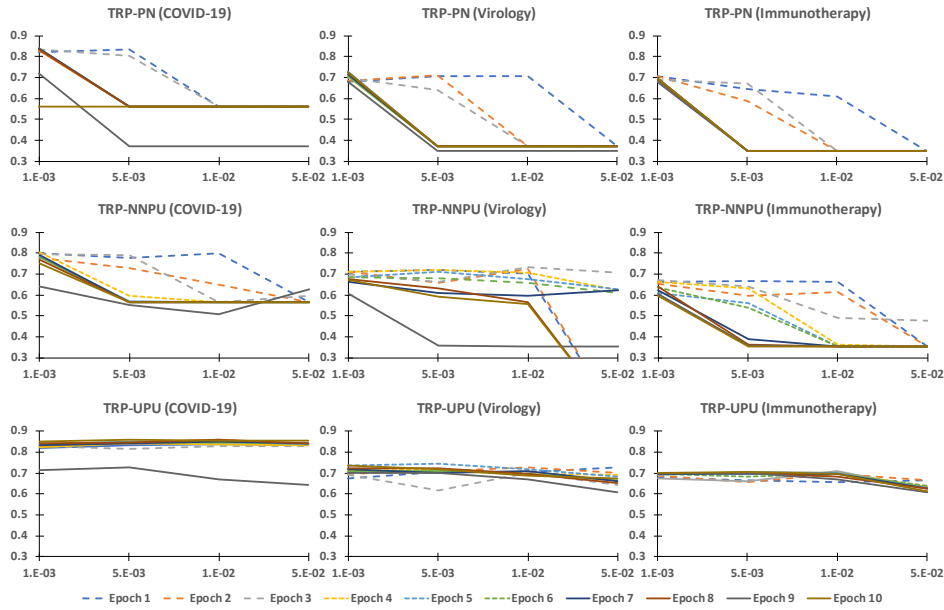

Figure 2: Stability comparison of TRP-PN, TRP-nnPU and TRP-uPU, showing the F1-S performance of the models (Y-axis) with different learning rates (X-axis) on 10 epochs.

## 4.4 Incremental Prediction

In Figure 3, we compare the performance of the top-performing PU learning methods on different year splits. We train **TRP** and other baseline methods on data until $t-1$ and evaluate its performance on predicting the testing pairs in $t$. It is expected to see performance gain over the incremental training process, as more and more data are used. We show F1-P due to the similar pattern on other metrics. We observe that the TRP models display an incremental learning curve across the three datasets and outperformed all other models.

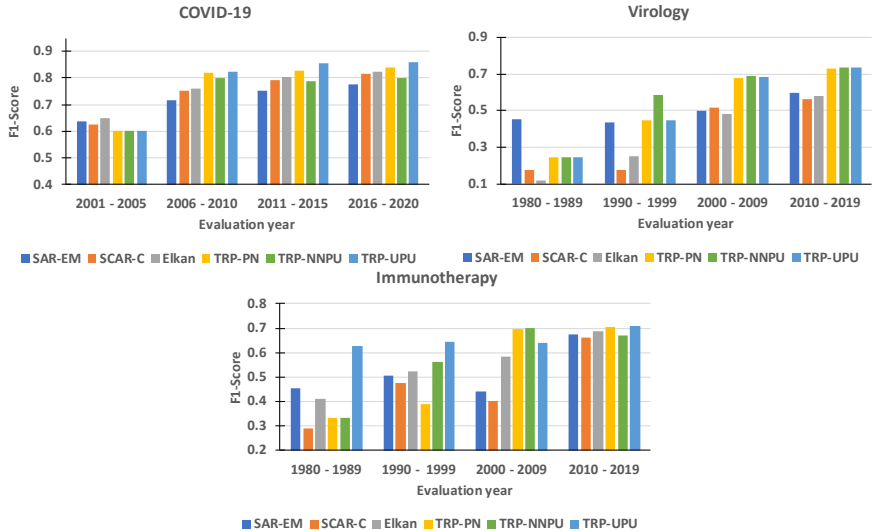

Figure 3: F1-P per year. The models are incrementally trained with data before the evaluation year.

## 4.5 Qualitative Analysis

We conduct qualitative analysis of the results obtained by TRP-uPU on the COVID-19 dataset. This investigation is to qualitatively check the meaningfulness of the paired terms, e.g., can term *covid-19* be paired meaningfully with other terms. We designed two evaluations. First, we set our training data until 2015, i.e., excluding the new terms in 2016-2020 in the COVID-19 graph, such as *covid-19, sars-cov-2*. The trained model then predicts the connectivity between *covid-19* as a new term and other terms, which can be also a new term or a term existing before 2015. Since new terms like *covid-19* were not in training graph, their term feature were initialized as defined in Section 4.1. The top predicted terms predicted to be connected with *covid-19* are shown in Table 3 top, with the verification in COVID-19 graph of 2016-2020. We notice that the top terms are truly relevant to *covid-19*, and we do observe their connection in the evaluation graph. For instance, *Cough*, *Fever*, *SARS*, *Hand* (washing of hands) were known to be relevant to *covid-19* at the time of writing this paper.

In the second evaluation, we trained the model on the full COVID-19 data ($\leq 2020$) and then predict to which terms *covid-19* will be connected, but they haven't been connect yet in the graph until 2020[1]. We show the results in Table 3 bottom, and verified the top ranked terms by manually searching the recent research articles online. We did find there exist discussions between *covid-19* and some top ranked terms, for example, [3] discusses how *covid-19* affected the market of *Chromium oxide* and [23] discusses about caring for people living with *Hepatitis B virus* during the *covid-19* spread.

## 4.6 Pair Embedding Visualization

We further analyze the node pair embedding learned by TRP-uPU on the COVID-19 data by visualizing them with *t-SNE* [34]. To have a clear visibility, we sample 800 pairs and visualize the learned embeddings in Figure 4. We denote with colors the observed labels in comparison with the predicted

Table 3: Top ranked terms predicted to be connected with term *covid-19*, trained until 2015 (the top table) and until 2020 (the bottom table). Verification of existence (Ex) was conducted in the graph in 2020 when trained until 2015, and by manual search otherwise. Sc is the predicted connectivity score.

| Terms | Sc | Ex | Terms | Sc | Ex | Terms | Sc | Ex | Terms | Sc | Ex |
|---|---|---|---|---|---|---|---|---|---|---|---|
| Leukocyte count | 0.98 | Yes | Air | 0.85 | Yes | Infection control | 0.96 | Yes | Serum | 0.84 | Yes |
| Fever | 0.94 | Yes | Lung | 0.81 | Yes | Population | 0.93 | Yes | Ventilation | 0.76 | Yes |
| Hand | 0.91 | Yes | SARs | 0.70 | Yes | Public health | 0.88 | Yes | Cough | 0.71 | Yes |

| Terms | Sc | Ex | Terms | Sc | Ex | Terms | Sc | Ex | Terms | Sc | Ex |
|---|---|---|---|---|---|---|---|---|---|---|---|
| Antibodies | 0.99 | Yes | Lymph | 0.99 | Yes | Tobacco | 0.99 | Yes | Adaptive immunity | 0.99 | Yes |
| A549 cells | 0.99 | Yes | White matter | 0.99 | Yes | Serum albumin | 0.99 | Yes | Allopurinol | 0.99 | Yes |
| Hepatitis b virus | 0.99 | No | Alkaline phosphatase | 0.99 | Yes | Macrophages | 0.99 | Yes | Liver function tests | 0.99 | Yes |
| Mycoplasma | 0.99 | Yes | Zinc | 0.99 | Yes | Bacteroides | 0.99 | No | Chromium dioxide | 0.96 | No |

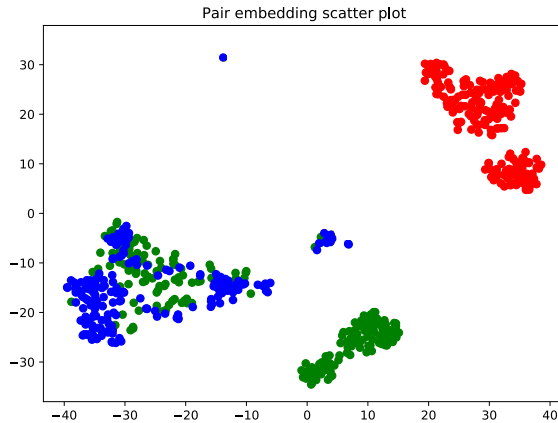

Figure 4: Pair embedding visualization. The blue color denotes the *true positive samples*, the red points are *unobserved negative*, the green points are *unobserved positive*.

labels. We observe that the *true positives* (observed in $G^T$ and correctly predicted as positives - blue) and *unobserved negatives* (not observed in $G^T$ and predicted as negatives - red) are further apart. This clear separation indicates that the learned $h$ appropriately grouped the *positive* and *negative (predicted)* pairs in distinct clusters. We also observe that the *unobserved positives* (not observed in $G^T$ but predicted as positives - green) and *true positives* are close. This supports our motivation behind conducting PU learning: the *unlabeled* samples are a mixture of *positive* and *negative* samples, rather than just *negative* samples. We observe that several *unobserved positives* are relationships like between *Tobacco* and *covid-19*. Although they are not connected in the graph we study, several articles have shown a link between terms [2, 46, 37].

## 5   Conclusion

In this paper, we propose *TRP* - a temporal risk estimation PU learning strategy for predicting the relationship between biomedical terms found in texts. TRP is shown with advantages on capturing the temporal evolution of the term-term relationship and minimizing the unbiased risk with a positive prior estimator based on variational inference. The quantitative experiments and analyses show that TRP outperforms several state-of-the-art PU learning methods. The qualitative analyses also show the effectiveness and usefulness of the proposed method. For the future work, we see opportunities like predicting the relationship strength between drugs and diseases (TRP for a regression task). We can also substitute the experimental compatibility of terms for the term co-occurrence used in this study.

## Acknowledgments and Disclosure of Funding

The research reported in this publication was supported by funding from the Computational Bioscience Research Center (CBRC), King Abdullah University of Science and Technology (KAUST), under award number URF/1/1976-31-01, and NSFC No 61828302. Additional revenue related to this work: student internship at Sony Computer Science Laboratories Inc. We would like to acknowledge the great contribution of Sucheendra K. PalaniapPalaniappanpan and The Systems Biology institute to this work for the initial problem definition and data collection.

## 6   Broader Impact

TRP can be adopted to a wide range of applications involving node pairs in a graph structure. For instance, the prediction of relationships or similarities between two social beings, the prediction of items that should be purchased together, the discovery of compatibility between drugs and diseases, and many more. Our proposed model can be used to capture and analyze the temporal relationship of node pairs in an incremental dynamic graph. Besides, it is especially useful when only samples of a given class (e.g., positive) are available, but it is uncertain whether the unlabeled samples are positive or negative. To be aligned with this fact, TRP treats the unlabeled data as a mixture of negative and positive data samples, rather than all be negative. Thus TRP is a flexible classification model learned from the positive and unlabeled data.

While there could be several applications of our proposed model, we focus on the automatic biomedical hypothesis generation (HG) task, which refers to the discovery of meaningful implicit connections between biomedical terms. The use of HG systems has many benefits, such as a faster understanding of relationships between biomedical terms like viruses, drugs, and symptoms, which is essential in the fight against diseases. With the use of HG systems, new hypotheses with minimum uncertainty about undiscovered knowledge can be made from already published scholarly literature. Scientific research and discovery is a continuous process. Hence, our proposed model can be used to predict pairwise relationships when it is not enough to know with whom the items are related, but also learn how the connections have been formed (in a dynamic process).

However, there are some potential risks of hypothesis generation from biomedical papers. 1) Publications might be faulty (with faulty/wrong results), which can result in a bad estimate of future relationships. However, this is a challenging problem as even experts in the field might be misled by the faulty results. 2) The access to full publication text (or even abstracts) is not readily available, hence leading to a lack of enough data for a good understanding of the studied terms, and then inaccurate $h$ in generation performance. 3) It is hard to interpret and explain the learning process, for example, the learned embedding vectors are relevant to which term features, the contribution of neighboring terms in the dynamic evolution process. 4) For validating the future relationships, there is often a need for background knowledge or a biologist to evaluate the prediction.

Scientific discovery is often to explore the new nontraditional paths. PU learning lifts the restriction on undiscovered relations, keeping them under investigation for the probability of being positive, rather than denying all the unobserved relations as negative. This is the key value of our work in this paper.

## Footnotes

[1] Dataset used in this analysis was downloaded in early March 2020 from `https://www.semanticscholar.org/cord19/download`

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
