[Supplementary Material]

# Supplementary Material: Temporal Positive-unlabeled Learning for Automatic Biomedical Hypothesis Generation

## 1 Data Preparation

### 1.1 Graph Construction

Given a dataset of scholarly publications (e.g., from PubMed), we extract and categorize the terms in the documents defined on a set of UMLS [18] and MeSH [1] terms. In this study, we use data from the pubmed database of March 2019 and Semantic scholar COVID-19 dataset [2] collected in March 2020. Each UMLS term belongs to one of three categories, namely: 1) *Genes*, 2) *Chemicals*, and 3) *Diseases*. We construct a network $G = \{V, E\}$, where $V$ is the set of nodes corresponding to the biomedical terms. The relationship $E$ represents the close co-occurrence of the two terms in literature. To be specific, an edge in $E$ connects two nodes if the two corresponding terms are mentioned together in the same title, abstract, or paragraph of a paper[1].

Next, we split the obtained network using year windows, thereby, obtaining a sequence of temporal graphlets $G = \{G^1, G^2, ..., G^T\}$. As defined in the Introduction Section 1 of the main paper, this graphlet sequence encapsulates the temporal evolution of node pair relationships. Since the node terms belong to several categories (e.g., drugs and diseases), the graph $G^t = \{V^t, E^t, x^t\}$ is, in fact, a dynamic heterogeneous attributed graph, with incremental nodes $V^1 \subseteq V^2, ..., \subseteq V^T$ and edges $E^1 \subseteq E^2, ..., \subseteq E^T$. The node attribute $x^t$ is composed of the term description when available, and the term contexts, which are the aggregation of sentences encompassing the mention of the terms in the documents. We use the texts from the publication titles, abstracts, and full-text paragraphs when available. The node attributes vary per time window due to the increase in the number of publications.

### 1.2 Positive Samples Construction

For each time step $t$, we construct only the node pairs of positive samples, since the negative pairs are uncertain. The positive node pairs are identified based on the graph observed at the next time step. Denote $a^{ij} = < v_i, v_j >$ a node pair consisting of nodes $v_i$ and $v_j$. As shown in Figure 1 of this supplementary document, the node pair $a^{ij}$ at time step $t$ is assigned a positive class $+1$ if a connection between node $v_i$ and $v_j$ is observed in graph $G^{t+1}$ (i.e., $s_t^{ij} = +1 \iff e(v_i, v_j) \in E^{t+1}$). Otherwise, the node pair $a^{ij}$ remains as unlabeled.

Since we consider the insertion only graphlets sequence, the graph size of the graphlets grows proportionally with the increase in time step. Therefore, the use of all possible pairs for training becomes more computationally expensive and less feasible in application. In this study for large graphs, the notion of a node pair set is defined as a sampled subset of all possible node pairs. This sample is drawn uniformly for each time step $t$.

Figure 1: An illustration of the graph sequence $G = \{G^1, G^2, ..., G^T\}$ with the sequential positive-unlabeled supervision in $A = \{A^1, A^2, ..., A^T\}$ (T=4 here). As the graph grows with more nodes and more connections, $A^t$ is constructed from the graph $G^{t+1}$. One unit at $i, j$ of $A^t$ is filled by 1 (a positive relation) if node $i$ and $j$ are connected in $G^{t+1}$. The unit is filled by a 0 (an unknown relation) if node $i$ and $j$ are NOT connected in $G^{t+1}$. During training, since we do not observe the future, the node pairs in $A^T$ are equivalent to those in $A^{T-1}$. The difference between them in the colored units are only observed in testing. In the training process, $G^t$ and $A^t$ ($t = 1...T$) form training samples of node pairs with positive or unknown labels. When testing, we aim to predict the unobserved pair relationships (colored tiles), in $A^T$.

## 2 Neighborhood Aggregation

The neighborhood aggregation is handled by the aggregator network $f_G(.; \theta_G)$. This network uses GraphSAGE at its core to aggregate each node's information in a given node pair $a^{ij} = <v_i, v_j>$ to obtain a concise representation for them. For each node $v$ in the pair, the aggregation network takes as input the current node feature $x_v^t$ as well as the neighborhood information, which includes the node features of the sampled node neighbors $x_{Nr(v)}^t$. Given a maximum neighboorhood layer $M$ to consider for information aggregation; at each aggregation step $m$, the representation vectors of neighbors $\{\Gamma_u^{m-1}, \forall u \in Nr(v)\}$ at iteration $m-1$ are aggregated into a single vector $\Gamma_v^m$ at iteration $m$. Several aggregation techniques for the neighborhood aggregation are proposed in [14]. At the initial aggregation step $m = 0$, node vector $\Gamma_v^0$ is the input node attribute (i.e., $\Gamma_v^0 = x_v^t$). After the neighborhood aggregation steps, the final representation $z_v^t = \Gamma_v^M$. The performance of aggregators often depends on the property of the applied graph [14]. We evaluate different aggregators and report the best.

**Neighborhood Definition**. Following the principle of [14], to keep the computational footprint to a minimum, we work on a fix-size sample set of node neighbors instead of the full neighborhood nodes. Hence the notion of node neighbors $Nr(v)$ is defined as a fix-size sample of the full node neighborhood $\{u \in V : (u, v) \in E\}$. This sample is drawn uniformly at each iteration, thereby reducing the time and memory complexity. With the sampling strategy, the memory and time complexity per node aggregation step is fixed at $\mathcal{O}\big(\prod_{m=1}^M S_m\big)$, where $S_m$ is the neighborhood sample size at layer $m$, and $M$ is the maximum layer considered (i.e., up to $M$-hop neighbors).

## 3 Dataset

In this project, each dataset contains the title and abstract of papers published in the biomedical fields. To evaluate the model's adaptivity in different scientific domains, we construct three graphs from papers on *COVID-19*, *Immunotherapy*, and *Virology*.

The graph statistics are shown in Table 1 of the main paper. To set up the training and testing data, we split the graph by a 10-year interval starting from 1949 (i.e., $\{\leq 1949\}, \{1950 - 1959\}, \ldots, \{2010 - 2019\}$) for the virology and immunotherapy datasets. Due to the novelty of the COVID-19 virus, we split the graph by a 5-year interval starting from 1995 (i.e., $\{\leq 1995\}, \{1995 - 2000\}, \ldots, \{2010 - 2015\}$). We use year splits of $\leq 2009$ ($\{G^1, G^2, ..., G^7\}$) for training, and the final split $2010 - 2019$ for testing on the virology and immunotherapy datasets.

We use year splits of $\leq 2015$ ($\{G^1, G^2, ..., G^5\}$) for training, and the final split $2015 - 2020$ for testing on the COVID-19 datasets.

At each $t$, for a given node (a biomedical term), we extract its term description and context (sentences encompassing the term in literature). The term description and contexts are respectively converted to a 300-dimensional feature vector by applying the latent semantic analysis (LSI) method on the document-term matrix features. The missing term and context attributes are completed with zero vectors. At each time $t$, the context features are updated with the new information about them in discoveries, and publications.

# 4 Experimental Setup

## 4.1 Baselines in Experiments

We use node2vec [13] to learn the graph structure feature and concatenate it with the text attributes to obtain an enriched node representation. In our link prediction task, we concatenate the embeddings of each node pair together as the final features.

When conducting the baseline experiments, we reweight the unlabeled examples following the instructions from [11] for Elkan's baseline. As for SAR-EM [5], SCAR-C [5], SCAR-KM2 [23], SCAR-TIcE [4], we randomly select 30 features from the embedding features, which are used to calculate propensity score. For other hyper-parameters, we follow the same setting as their paper. However, due to training SAR-EM model is very time-consuming, and the result is very unstable, we limit the expectation-maximization iteration to 10,30,300, and we finally select the best performance among them.

## 4.2 TRP Model

In all our experiments, we treat the graph to be undirected and set the hidden dimensions to $d = 128$. For each neural network-based model, we performed a grid search over the learning rate $lr = \{1e^{-2}, 5e^{-3}, 1e^{-3}, 5e^{-2}\}$, on the Virology and Immunotherapy datasets from 1944 to 1999, and from 1950 to 2010 for the COVID-19 dataset. The best parameters per model from the grid search are then used in all experiments. The TRP models are trained with a parameter set ($d = 128$, $S_1 = 20$, and $S_2 = 10$), where $S_1, S_2$ are the neighborhood sample size for the one-hot and two-hop neighborhood aggregation respectively. We implement TRP on Python, using the Tensorflow library. Each GPU based experiment was conducted on an Nvidia 1080TI GPU. The code will be publicly available upon the acceptance of the work.

# 5 Evidence for Supporting the Discovered Pairs

In this section, we provide evidences supporting the connectivity prediction between COVID-19 and the terms in Table 3 of the main paper. Note that the cited reference papers as evidence here were not present in our training and testing graphs.

**Anti-bodies – COVID-19.** The relationship between antibodies and the COVID-19 is well known, and several articles have been published linking the two terms together. The relationship is mainly seen in articles about the research and development of vaccines.

**A549 cells – COVID-19.** There are several very recent studies on the effects of COVID-19 on the A549 cells. Specifically, the capacity of COVID-19 to infect and replicate in A549 cells [15, 6, 8].

**Mycoplasma – COVID-19.** Several articles studied the effects of coinfection of Mycoplasma and COVID-19 and their correlation [19, 12].

**White matter – COVID-19.** White matter is the parts of the brain that connect brain cells to each other. Brun et al. [7] studied and analyzed the effects of the COVID-19 virus on the neurological functions of the brain.

**Zinc – COVID-19.** The effect of zinc on common colds, mostly caused by rhinoviruses, has been studied. Although the novel coronavirus that causes COVID-19 is not the same type of coronavirus that causes common colds, several studies [9, 16] have been made on the effect of zinc supplements on COVID-19 virus.

**Tobacco – COVID-19.** Some researchers have studied and analyzed the effect of tobacco usage with COVID-19 [3, 24]. Another research is the Cotiana Project [20], which studies the potential use of tobacco for vaccine production.

**Macrophages – COVID-19.** Macrophages are a population of innate immune cells that sense and respond to microbial threats by producing inflammatory molecules that eliminate pathogens and promote tissue repair [17]. Several recent studies have shown the effects of Macrophages on COVID-19 [22, 17].

**Adaptive immunity – COVID-19.** Adaptive immunity is an immunity that occurs after exposure to an antigen either from a pathogen or a vaccination. Several works have studied the availability and duration of adaptive immunity after a patient has been exposed to the COVID-19 virus [21, 10].

## Footnotes

[1]A mention can have a positive or negative connotation. We consider any kind of mention as a relationship, regardless of positive or negative. We leave the study of edge polarity for future investigation.