[Reviews · NeurIPS 2020]

Review 1

Summary and Contributions: This paper proposes to use temporal PU learning to tackle the hypothesis generation (HG) problem. Specifically, the authors formulate the HG problem as future connectivity prediction on a dynamic attributed graph. The authors claim that the experiments on COVID-19 datasets validate the effectiveness of the proposed model.

Strengths: 1. The idea of this paper is very novel, which combines PU learning, GRU, and graphSAGE. 2. The application to COVID-19 is much appreciated.

Weaknesses: 1. The writing and presentation of this paper can be further improved. 2. The proposed method is not compared with the SOTA methods, and some important baselines are missing. 3. The codes and datasets are not provided, which decreases the reproducibility. I understand that the data may be confidential, but the authors should at least provide the codes for their algorithm.

Correctness: Maybe correct.

Clarity: Can be improved.

Relation to Prior Work: Some important prior works are missing.

Reproducibility: No

Additional Feedback: 1. In Introduction, when I read the sentence “When two terms co-occurred at time t in scientific discovery …”, in fact I’m not pretty sure what are the “terms” in this paper. The authors should explain this notion in advance in a clearer way. 2. The authors claim that they proposed a variational inference model to estimate the positive prior. In fact, the main idea is to minimize the difference between two distributions. Such idea has already been presented in “Class-prior Estimation for Learning from Positive andUnlabeled Data” (ACML 15). The main difference is that ACML paper uses L1-distance while this paper uses KL divergence. Therefore, the ACML method should be discussed and compared. 3. In related work of PU learning, there are other important works that considered the unlabeled examples as negatives with label noise, such as “Loss decomposition and centroid estimation for positive and unlabeled learning” (PAMI 19) and “Positive and Unlabeled Learning via Loss Decomposition and Centroid Estimation” (IJCAI 18). These two papers can be cited. 4. The compared PU baselines are a bit weak. The authors claim that “The used SOTA PU learning methods include [14] …” However, [14] is published in 2008 and is not SOTA. SOTA includes nPU, nnPU, LDCE, and some GAN-based PU models. Therefore, the comparisons with PU learning methods should be improved. 5. The texts in figures are too small and unclear, such as Fig. 1 and Fig. 2. 6. The presentation of this paper can be further improved, for example “The idea is to find variational distribution variables theta* that minimize the Kullback-Leibler (KL) divergence” should be “The idea is to find variational distribution variables theta* that minimizes the Kullback-Leibler (KL) divergence”. Generally, I feel that this is a borderline paper. However, considering that this paper contains some publishable results, I currently give a positive score on this paper. -----------------------------update after rebuttal---------------------------- I thank the authors for addressing my concerns. I hope the experimental part of this paper can be enhanced if this paper is finally accepted. Besides, the release of codes and datasets as promised by the authors is also appreciated.


Review 2

Summary and Contributions: The paper designs a novel algorithm named TPR in PU-Learning to predict future connectivity on a dynamic attributed graph (HG problem). The algorithm can be divided into two parts. The first part is to transform the HG problem into a PU learning problem, so that the unbiased risk estimator based PU methods can be applied. The second part is about estimation of the positive prior also by optimizing an objective function related to ELBO.

Strengths: The ideas and process of the algorithm are sound and clear, and the proposed methods have comparably high accuracy on ground-truth datasets. The major contribution of this paper is to transform a practically important problem, mecial HG problem, into a PU learning problem. It shows the potential of the PU learning in real-world applications besides image classification.

Weaknesses: 1) The class prior estimation is an intractable problem for PU learning, and it is hard to identify the quantity from data without the assumption of irreducibility (the negative distribution cannot be a mixture that contains positive distribution). The authors seem to avoid this problem by using the Guassian mixture model with two components, but it leads to another problem: Is the GMM suitable for the data? 2) This is related to the above one. Can authors show (at least by numerical results) that the class prior is well estimated in experiments? I am curious how the ratio of positive dataset to unlabeled dataset influences effectiveness of the model. 3) It is interesting to see that nnPU is worse than uPU in experiments. I think more analysis is required, because the to optimal PU classifier must satisfy the "the non-negative restriction of the risk estimation" according to the theoretical analysis. ==== Update after rebuttal: I thank the authors for addressing the main points. But more details on the estimation of class prior is still required.

Correctness: Almost correct except the part mentioned in "Weaknesses"

Clarity: Yes

Relation to Prior Work: Yes

Reproducibility: Yes

Additional Feedback:


Review 3

Summary and Contributions: This paper is aimed at addressing hypothesis generation problem. It considers the HG problem as connectivity prediction by capturing the features on a temporally dynamic graph with positive-unlabeled learning. Also, the author proposes a variational inference model to estimate positive prior to help node embedding learning. I have read the rebuttal.

Strengths: (1) The author proposes the Temporal Relationship Predictor model to calculate the future connection score for term pairs based on PU learning framework. It’s the first application of PU learning on the HG problem and dynamic graph. (2) To acquire a more reliable positive prior, the authors proposes a variational inference method, treating the learned pairs embedding with Gaussian mixture distribution and minimize the KL divergence to estimate the GMM model parameters \beta. (3) Experimental results show the effectiveness of TRP method, achieve the SOTA result in PU learning. The authors also give a detailed analysis and visualization for the result.

Weaknesses: (1) The introduction to L^E in eq(7) can be more clear. (2) It would be better to conduct some ablation studies to show the effectiveness of prior estimate.

Correctness: yes

Clarity: yes

Relation to Prior Work: yes

Reproducibility: No

Additional Feedback: Overall, the paper is well written, and model structure and training details are clearly presented. Questions: 1. Would you release the datasets?


Review 4

Summary and Contributions: The paper contributes a method for modeling the evolution of connections in a graph that considers links that are unobserved so far as unlabeled rather than negative. It is applied to a hypothesis generation problem by modeling the cooccurrence of biomedical terms in paper titles and abstracts over the last 75 years.

Strengths: The treatment of links that are unobserved so far as unlabeled rather than assuming they are negative/absent makes a lot of sense. The approach of modeling the temporal evolution of the graph also seems advantageous.

Weaknesses: The exposition of the methodology is dense and hard to follow (Section 3). Is there a way to provide confidence intervals on the values in Table 2?

Correctness: I would like to understand the part about estimating p(y=1) better. I was under the impression that this quantity was unidentifiable from positive-unlabeled data without strong assumptions. Does the SCAR assumption enable estimation? Does it rely on other assumptions as well?

Clarity: There is substantial room for improvement in clarity. - The text in all figures is way too small to be reasonably readable. - The paper would benefit from another editing pass for English grammar. Minor points: - h is undefined at first use. - GRU is not spelled out.

Relation to Prior Work: Seems reasonable -- but see questions about estimating p(y=1).

Reproducibility: Yes

Additional Feedback: I have some skepticism about the utility of this approach for hypothesis generation (HG). I see in the results that some terms were linked at the last time step which truly did get added at the end of the time series. But how do you envision this informing scientific research? That if two terms are predicted to co-occur, it will inspire a new study? Are there examples (very generally) of HG leading to new scientific discoveries? Thanks to the authors for the feedback provided. I think these changes will improve the paper. I am changing my reproducibility score to "Yes" since the authors plan to release the code and datasets. I am changing my overall score to 6.

[Author Response · NeurIPS 2020]

| | COVID-19 | | | | Virology | | | | Immunotherapy | | | |
|---|---|---|---|---|---|---|---|---|---|---|---|---|
| | F1-S | F1-M | F1-P | LRAP | F1-S | F1-M | F1-P | LRAP | F1-S | F1-M | F1-P | LRAP |
| penL1-NNPU | 0.71 | 0.70 | 1.35 | 0.45 | 0.61 | 0.71 | 1.83 | 0.63 | 0.53 | 0.63 | 1.62 | 0.73 |
| TRP-NNPU | 0.80 | 0.82 | 1.68 | 0.89 | 0.73 | 0.82 | 2.38 | 0.83 | 0.67 | 0.78 | 2.18 | 0.76 |
| penL1-UPU | 0.85 | **0.88** | 1.86 | 0.89 | **0.74** | **0.83** | **2.45** | **0.81** | 0.70 | 0.81 | 2.33 | 0.72 |
| TRP-UPU | **0.86** | **0.88** | **1.88** | **0.91** | **0.74** | **0.83** | 2.38 | **0.81** | **0.71** | **0.82** | **2.35** | **0.77** |

We would like to thank the reviewers for the recognition of the novelty of our studied problem and our proposed solution,
the attention to our evaluation including a Covid-19 dataset, as well as their valuable suggestions and comments.

**Organization and Writing (ALL)**    We have gone through the paper multiple times to correct the grammatical and
structural mistakes in the paper. We also address the issues raised by the reviewers, which are: 1) explain all notations
before usage, 2) clearly denote the cost function $L^E$ in Eq(7), and 3) spell out all acronyms. Given that the page limit
for review submission is eight pages, the images were reduced to fit the limit. We will increase the fonts and image
sizes of the figures in the camera-ready version, which allows for an additional ninth page.

**Related works Citation and Comparison (R1)**    We have cited the suggested papers [**I-III**], and also compared with
the *pen-L1* method proposed in [**I**], for further evaluating the positive prior estimation in our TRP method. The results
presented in the table above show that our TRP outperforms or is comparable to the *pen-L1* method in both UPU and
NNPU setting. Thus, for positive prior estimation, it is more effective to use KL-divergence to estimate the parameters
of class mixture distribution, than to assign higher penalties by penalized L1 divergences (*pen-L1*) to class priors that
scale the positive distribution more than the total distribution. We will add this comparison in our paper.

**Prior Estimation Assumptions (R2,R4)**    We follow a basic and the most generic SCAR assumption in PU learning:
the labeled positive examples are chosen completely randomly from all positive examples. This means, for all positive
$x$, they have the same probability to be labeled (sampled as observed positive instances). This is the positive prior
probability $\pi_P = p(y = 1)$ to estimate, since there exist unobserved positive instances in the unlabeled set. Then we
follow the assumption of *positive subdomain/anchor set* in [5] and [**I**], which realistically assumes the distribution of
positive and negative may overlap, but the labeled positive form an anchor set in the positive subdomain. We thus use a
Gaussian mixture model (GMM) to fit the data distribution, which is made up of two components: positive and negative.
The estimated prior $\pi_P$ is the weight of the positive component. Thus, GMM is a suitable model under the *positive
subdomain/anchor set* assumption for the positive prior estimation.

**Exp Result and Discussion (ALL)**    (**R1**) In our experiments, we compared against recent SOTA methods that jointly
learn a PU classifier while estimating the class prior or propensity score. They are: Elkan [14] proposed in 2008,
SAR-EM and SCAR-C [7] proposed in 2019, SCAR-KM2 [35] proposed in 2016, and SCAR-TIcE [5] proposed in
2018. We now include *pen-L1* method proposed in 2015.
(**R2**) An increase in the number of positive samples improves the effectiveness of the model, as shown in Figure 3,
where the number of training instances increases with the introduction of labeled pairs observed in incremental years.
(**R2,R3**) It is hard to provide a detailed comparison of the prior estimates given our dataset since the true prior
is unknown in our real-world setting. Hence we evaluate the performance of the model as a whole based on the
classification performance metrics. The results presented in Table 2 in the paper and in the above table validate the
effectiveness of our prior estimation method, comparing to other prior estimation solutions.
(**R2**) We also found interesting that NNPU was worse than UPU in our experimental results. However, it is not
uncommon for UPU to outperform NNPU in evaluation with real-world datasets. Similar observations were found
in the results in [**I**] and [**II**]. In our case, we attribute this observation to the joint optimization of the loss from the
classifier and the prior estimation. In the loss of UPU (Eq. (3)), $\pi_P$ played in weighting both $\hat{\mathcal{R}}_P^+(f)$ and $\hat{\mathcal{R}}_P^-(f)$.
However, in the loss of NNPU (Eq. (4)), $\pi_P$ only weighted $\hat{\mathcal{R}}_P^+(f)$ when $\hat{\mathcal{R}}_U^- - \pi_P\hat{\mathcal{R}}_P^-(f)$ is negative. In real-world
applications, especially when the true prior is unknown, the loss selection affects the estimation of $\pi_P$, and thus the
final classification results.
(**R1,R3**) The full **code** and **datasets** will be provided upon acceptance.
(**R4**) We have included confidence intervals of TRP performance in Table 2.

**Usefulness of HG (R4)**    Hypothesis generation can provide domain experts clues about the relevant concepts to
explore. Following these clues, domain experts can retrieve the documents, including the relevant concepts, and then
investigate the hypothesis's usefulness. For instance, *Schizophrenia* and *Calcium-Independent Phospholipase A2* were
independently studied in different papers in 1997 and 1995. They were connected in 1998 [**IV**] because they had
*oxidative stress* as a common factor. The *Five Golden Test Cases* are also often used in the qualitative evaluation of
hypothesis generation papers [41,25,45]. Our next target is to develop a system that can generate a sentence for the
generated hypothesis, including the context explaining why and how they are relevant. This requires to enrich the
currently studied graph with the contextual information for the nodes and edges. The system will help to new scientific
discoveries in a more explainable way.

**Reference**

[I] Christoffel et al., "Class-prior estimation for learning from positive and unlabeled data." ACML. 2016.
[II] Gong et al., "Loss decomposition and centroid estimation for positive and unlabeled learning." PAMI. 2019.
[III] Shi et al., "Positive and Unlabeled Learning via Loss Decomposition and Centroid Estimation." IJCAI. 2018.
[IV] Smalheiser and Swanson, "Calcium-independent phospholipase A2 and schizophrenia." Archives of General Psychiatry. 1998.


[Meta-Review · NeurIPS 2020]

This paper considers temporal positive-unlabeled learning to predict future connectivity on a dynamic attributed graph (i.e., the hypothesis generation problem). The major contribution is to transform the problem of interest into PU learning. The proposed method is shown to be applied on COVID-19 data! The clarity and novelty are clearly above the bar of NeurIPS. While the reviewers had some concerns on the experiments, the authors did a particularly good job in their rebuttal. Thus, all of us have agreed to accept this paper for publication! I have two additional comments. First of all, the learning objective used in the paper is not originally from [26]. Indeed, it is from [M. C. du Plessis, G. Niu, and M. Sugiyama. Convex formulation for learning from positive and unlabeled data. ICML 2015]; see the unnumbered equation above Eq. (3) of this paper. This paper is mostly cited together with [12] and [26], especially this is what you are really using in 3.1 Model Design, but it seems that you missed this reference. Then, as pointed out by R2, the irreducibility assumption is an issue of PU class-prior estimation. There is a paper on arxiv entitled "towards mixture proportion estimation without irreducibility" which is as far as I know the latest result and can be used to estimate PU class prior when the irreducibility assumption fails to hold in practice. PS, unbiased PU should be shorten to uPU and non-negative PU should be shorten to nnPU (at least in the nnPU paper which is the first paper using these two names). UPU and NNPU look like two new problem settings named unlabeled-positive-unlabeled and negative-negative-positive-unlabeled classification... PS2, for the name "Marthinus Christoffel du Plessis" (who is the first author of [9,12,13]), Christoffel is his middle name but not last name. The short version should be "M. C. du Plessis" as shown above or "du Plessis, M. C." where du is small even if at the beginning of a sentence (as a rule of Afrikaans).